

# European summer precipitation

Birthe Marie Steensen[1], Gunnar Myhre[1], Racheal Byrom[1], Ada Gjermundsen[2,3], Caroline Jouan[1], Camilla Weum Stjern[1]

[1]Center for International Climate Research (CICER), Oslo, 0318, Norway
[2]Section for Meteorology and Ocean Sciences, University of Oslo, Oslo, 0318, Norway
[3]Norwegian Meteorological Institute, Oslo, 0313, Norway

*Correspondence to*: Birthe Marie Steensen (birthe.steensen@cicero.oslo.no)

**Abstract.** Observations indicate that parts of Europe have experienced summer drying over recent decades. There is, however, substantial uncertainty in terms of precipitation changes projected by global climate models, underscoring the need for
improved understanding to guide adaptation. Here, we analyse changes in European summer precipitation with global warming in 35 CMIP6 Earth system Models using a regional atmospheric energy budget framework. Over the historical period we compare these models to two reanalysis datasets, ERA5 and MERRA-2, as well as the E-OBS daily gridded observational dataset. Although ERA5 and MERRA-2 compares well to observed precipitation, both reanalysis datasets exhibit large discrepancies with the CMIP6 models when evaluating other terms in the atmospheric energy. In future projections, the model
spread is increased compared to historical diversity for the change in precipitation, sensible heat and the dry static energy flux divergence, highlighting the uncertainties in the magnitude of these terms. Nevertheless, the models show consistent agreement on the projection of summer drying over continental Europe with anthropogenic forcing over time.

## 1 Introduction

The hydrological cycle and precipitation patterns are changing with global warming (Allan *et al.*, 2020; Douville *et al.*, 2021). For Europe, both observations and Earth System Models (ESMs) indicate a precipitation change with drying over the continent and Mediterranean during the summer months (Rowell and Jones, 2006; Seager et al., 2019). There is, however, still a large spread between Coupled Model Intercomparison Project Phase 6 (CMIP6) models for future projections of precipitation over Europe in summer (IPCC, 2021), although the projected range in CMIP6 is reduced compared to CMIP5 (Palmer, Booth and
McSweeney, 2021). The Mediterranean region has especially experienced a drying since the 1980s and is characterized as a climate hotspot, due to the anticipated continuation of warming and drying (Lionello and Scarascia, 2018). There has also been an increase in summer droughts over continental Europe since 1901, except over Northern Europe (Hänsel *et al.*, 2022). Increasing temperatures further intensifies drought periods (Spinoni *et al.*, 2015, 2020). The trend in extreme precipitation (Rx1day; the annual maximum 1 day precipitation) also shows a decrease over stations in the Mediterranean (Sun *et al.*, 2021),
while there is a significant increasing Rx1day trend in Northern Europe with global warming (Dyrrdal *et al.*, 2021).





The atmospheric energy budget represents the balance between incoming energy and energy lost by the atmosphere, implying that variations in precipitation are linked to changes in atmospheric radiative cooling and surface sensible heat (SH) fluxes (O'Gorman et al., 2012). While the Clausius-Clapeyron relation states that for each degree warming the atmosphere can contain 7% more water vapor until saturation, precipitation and evapotranspiration are constrained by the atmospheric energy balance

at 2-3 % increase per degree warming (Allen and Ingram, 2002; Allan *et al.*, 2020).Analyzing precipitation changes through energetic constraints in different ESMs is important to understand the modelled differences in climate projections (Muller and O'Gorman, 2011). For example, incoming and outgoing radiation is dependent on clouds, aerosol scattering and absorption, water vapor absorption and albedo (Wild, 2020; Loeb *et al.*, 2022; Wild and Bosilovich, 2024). An identified improvement from ESMs contributing to CMIP5 and ESMs contributing to CMIP6 is the adjustment of the too large shortwave (SW)

downward radiation due to a lack of absorption in a cloud free atmosphere (DeAngelis *et al.*, 2015; Pendergrass, 2020). A better representation of the water vapor continuum over the model generations has also improved the downward longwave radiation (LW) (Wild, 2020). Another example is that the higher Equilibrium Climate Sensitivity (ECS) in CMIP6 models compared to CMIP5 have been attributed to radiation changes caused by a decrease in extratropical low clouds (Zelinka *et al.*, 2020). However, these radiative adjustments have been evaluated on the global annual mean scale for the atmospheric energy

budget, and their regional and seasonal effects remain unclear. This is crucial for understanding model trends in regional precipitation, which often have greater societal impacts than global-scale changes. In this article we will study the changes in the atmospheric energy budget for summer precipitation over Europe.

The SH term also exhibits a large spread among the CMIP6 models, reflecting uncertainties in quantifying the different terms (e.g. turbulence, winds, temperature) included in the SH calculations. Consequently, the SH is one of the most uncertain terms

in the global energy budget (Wild, 2020; Myhre *et al.*, 2024).

Even though reanalysis datasets are constrained by observations, they still show a large spread in estimates of the global energy budget components, including both radiation and SH (Irving *et al.*, 2021; Wild and Bosilovich, 2024).

Changes in precipitation are of high societal importance, with water availability having crucial impacts for agriculture and energy production etc. Changes due to global warming over highly populated areas like Europe are therefore important to

understand. In this study we will look at the difference in the energy balance terms between two historical time periods for 35 CMIP6 models and compare the differences to ERA5 and MERRA-2 reanalysis, as well as the precipitation difference for a gridded observational dataset E-OBS. Further, we investigate future projections for two Shared Socioeconomic Pathways (SSPs) for mid-century and at the end of the century. The following section outlines the methods used in study, the subsequent sections present the results following a discussion and conclusions in the end.




## 2. Data and Methods

### 2.1 Data

There are 35 CMIP6 models (Eyring *et al.*, 2016) used in this study, listed in Table B1. Since the study is focused on the spread between the models, we weigh each model the same even though some of the models are from the same modelling community or share the same physics and parameterizations. Historical forcings for the model simulations are based on observations of greenhouse gases, land use, solar forcing, volcanic emissions etc. and cover the period 1850-2014 (Eyring *et al.*, 2016). For future projections (2015-2100) we use a high greenhouse gas scenario SSP5-5.8 and a medium forcing scenario SSP2-4.5 (O'Neill *et al.*, 2016). To compare models over the historical period (1980-2023), we extend the historical simulations using SSP5-5.8 from 2015.

Reanalysis data from ERA5 (Hersbach *et al.*, 2020) and MERRA-2 (Gelaro *et al.*, 2017) are compared with the CMIP6 models. ERA5 has a resolution of 0.25 degrees for latitude and longitude, while MERRA-2 has a coarser resolution of 0.5 degrees in latitude and 0.625 in the longitude. MERRA-2 includes aerosol data assimilation, while ERA5 uses aerosol, greenhouse gases, ozone fields and total solar irradiance from CMIP5.

We also compare precipitation with gridded daily precipitation observations from E-OBS with 0.1-degree resolution (Cornes *et al.*, 2018). While ERA5 does not assimilate gauge data, and is therefore independent of precipitation observations, MERRA-2 corrects the precipitation from observationally based precipitation products.

### 2.2 Regional energy balance

The atmospheric energy balance is true only in a global-mean sense, and to get the regional atmospheric energy budget, a column integrated dry static energy flux divergence (H) needs to be included (Muller and O'Gorman, 2011). The precipitation (P) response to a change in climate ($\delta$) can be expressed as:

$$L_c\delta P = \delta SW + \delta LW - \delta SH + \delta H \tag{1}$$

Here, the latent heat of condensation $L_c$ converts precipitation change so that all terms are expressed in W m$^{-2}$. An increase in temperature due to increased greenhouse gases will increase longwave (LW) radiative cooling making a positive contribution to precipitation change (O'Gorman *et al.*, 2012). Warming also causes an increase in water vapor concentration that in turn causes increased absorption of shortwave (SW) radiation and a heating of the atmosphere, which has a negative contribution to precipitation. In addition, aerosols can both scatter (e.g., sulphate) and absorb (e.g., black carbon) SW radiation depending on their composition and altitude (DeAngelis *et al.*, 2015; Pendergrass, 2020; Wild, 2020). (DeAngelis *et al.*, 2015; Pendergrass, 2020; Wild, 2020). Clouds have a large impact on the atmospheric energy budget with direct radiative effects and feedback mechanisms, clouds absorb or scatter incoming SW radiation and outgoing LW radiation. The physical representation and geographical location of clouds is important for whether they contribute to radiative cooling or heating found in both observations and models (Zelinka *et al.*, 2020; Gjermundsen *et al.*, 2021; Loeb *et al.*, 2024). SH depends on the



difference between the surface and the overlying air, turbulence and convection (Myhre , 2024). The H term is calculated as a residual and is driven by changes in dry static energy in addition to changes in horizontal and vertical winds.

To prevent long-term thermal drift in the pre-industrial control climate, the CMIP models are constrained to achieve a global-
mean, annual-mean net top-of-atmosphere (TOA) radiation balance, thereby maintaining the balance expressed in Eq. (1) (Muller & O'Gorman, 2011). Most of the CMIP6 models achieve approximate energy budget closure, and show an improvement compared to CMIP5 models (Irving *et al.*. There are several methods to achieve radiative balance (or close to radiative balance), some models stabilize at a level of imbalance (drift) in their spin-up simulations e.g. for ACCESS-CM2 and ACCESS-ESM1-5 they gradually stabilize around 0.35 Wm-2 and -0.25  Wm-2 , causing the models to have a continuous
ocean heat uptake or loss (Bi *et al.*, 2020; Ziehn *et al.*, 2020)). Others tune the parameters for AMIP (Atmospheric Model Intercomparison Project) simulations to keep conservation (Döscher *et al.* 2022).In this study, the H term is calculated as a residual and may therefore also include changes in energy storage (Muller and O'Gorman, 2011). Appendix A1 shows the H term from the NorESM2-MM model, from June, July and August (JJA) 2015-2020. The dry static energy flux divergence is calculated using data at 3-hour temporal resolution, interpolated from terrain-following coordinates to pressure levels and then
integrated in the vertical. With a top-of-the-atmosphere radiative imbalance of -0.065 Wm-2 (Seland *et al.*, 2020), the large-scale differences in H between the model calculations and the residual from monthly files are comparable, even though the residual term exhibits a smoother field. Although models do not achieve perfect radiative balance, comparing them with the atmospheric balance equation provides valuable insight into the causes of model spread in European precipitation and how the hydrological cycle responds to global warming.

Here, we evaluate the energy balance equation over Europe (-10 to 30 degrees West and 35 to 75 degrees North) and three sub regions; the Mediterranean (-10 to 30 degrees West, 35 to 48 degrees North), a Central European region (-5 to 15 degrees West and 48 to 66 degrees North) and a Northern European region (10 to 30 degrees West and 64 to 74 degrees North), see Figure 3b). Individual models are analysed on their native grid over land only. Multi-model means are constructed by interpolating model data to a common grid of 1 x 1 degree in latitude and longitude using bilinear interpolation. The results are presented
as the change in summer (JJA) averages between the reference period 1980–1999 and a recent period (2004–2023), which includes both reanalysis and observational datasets. For future projections, we show changes relative to the same reference period and two time periods, the mid-century (2040–2059) and the end-of-century (2080–2099).





# 3 Results

## 3.1 Evaluating energy budget terms between models, reanalyses and observations

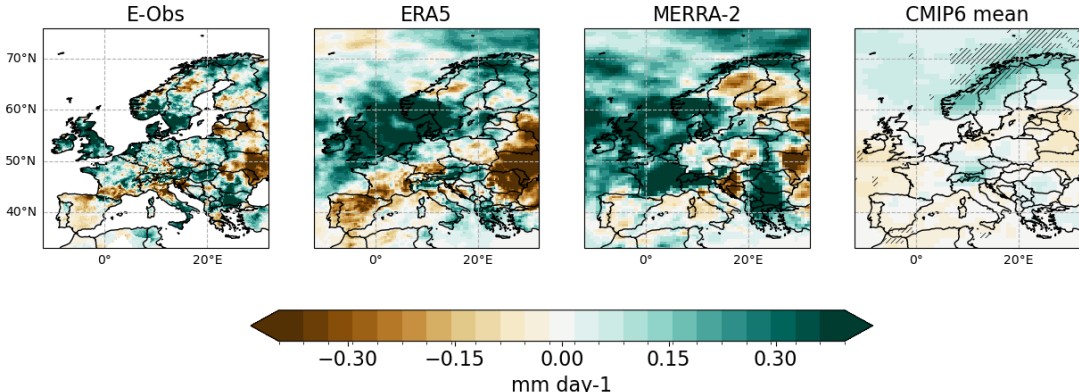


**Figure 1: Change in mean summer (JJA) precipitation between the two 20-year periods of 2004-2023 and 1980-1999 in E-OBS, ERA5, MERRA-2 and CMIP6 multi-model mean. For the model mean, hatching indicates where more than 75% of the models agree on the sign of change.**

Figure 1 shows precipitation change over Europe between the reference period (1980-1999) and the recent period (2004-2023),

where E-OBS show drying over the Mediterranean, except southern Italy and Greece. Ukraine, Belarus and the Baltic Countries also show a drying since the 1980s. The signal is less clear over Central Europe, and an increase over Ireland, the UK, Denmark and Southern Sweden and Norway is found. ERA5 reanalysis generally shows the same tendencies, although with a more pronounced drying over the western Mediterranean and Eastern Europe. Precipitation changes in MERRA-2 reanalysis show a larger area of increase over France and in eastern parts of the Mediterranean, however no large changes over

southern Italy.

The CMIP6 model mean shows much weaker precipitation changes and only shows model agreement (where more than 75% of the models agree on the sign of change) in the increase over the Nordics and the Alps. For the rest of Europe, the model mean either shows a small decrease or no change, with low model agreement. Figure 2 shows the model mean change in regional atmospheric energy budget for the same period, to investigate how the different terms evolve.




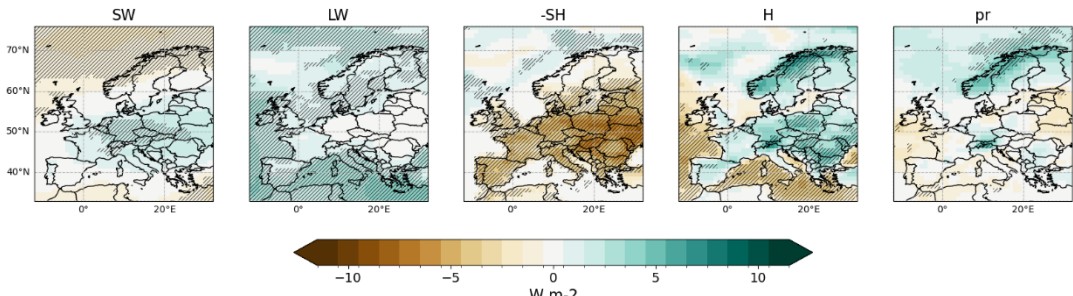

**Figure 2: Mean changes in components of the regional atmospheric energy budget contributing to precipitation, SW and LW radiative cooling, SH (shown as negative), and the mean residual H term, in addition to precipitation averaged over 35 CMIP6 models. The values represent the mean difference between the periods 1980–1999 and 2004–2023 over the summer months (JJA). Regions are hatched where more than 75% of the models agree on the sign of change.**

There is a larger model agreement for the SW, LW SH and H terms in the regional energy budget compared to the precipitation term. For SW atmospheric radiative cooling, over 75 % of the models show a decrease in the North Europe over ocean and land, as well as an increase in atmospheric radiative cooling over Central Europe. The CMIP6 models generally agree on an increase in LW atmospheric radiative cooling (positive contribution to precipitation change) over the oceans, with less agreement and change over Central Europe. An increase in SH contributes negatively to a change in precipitation (Eq. 1), therefore results are presented as a negative to reflect the resulting precipitation response. Models agree that SH over land contributes to reducing precipitation except over Northern Scandinavia. The change in dry static energy (H) shows a model mean where there is a positive contribution to precipitation over land and a negative contribution over the oceans. There is model agreement over the oceans south of 50 degrees north and only in scattered areas over land.

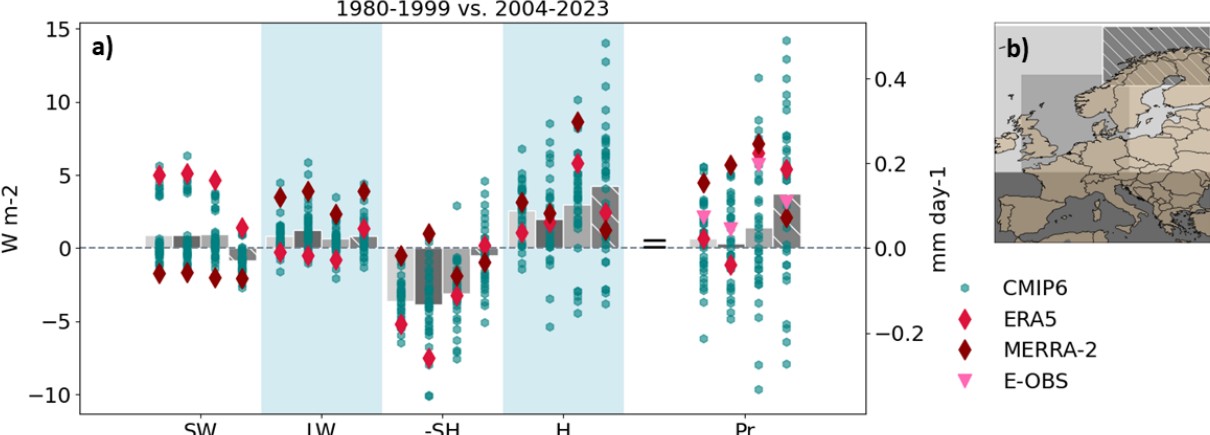

**Figure 3: a) The change in summer (JJA) regional atmospheric energy budget over land between the two 20-year periods for the CMIP6 models (green circles), ERA5 data (red diamonds), MERRA-2 (brown diamonds) and precipitation for E-OBS (pink triangle), over the four regions from left to right: Europe – light grey bar, Mediterranean – dark grey bar, Central Europe – grey**



**bar and Northern Europe – hatched dark-grey bar. The bars show the CMIP6 model mean over each region with the green circles representing results from individual CMIP6 models. b) Map showing the extent of each region in the same greys as their respective bars – recall that land regions only are included in the regional averages.**

To further study the spread in the models and how they compare to the reanalysis and observational data over Europe, the fluxes are spatially averaged over land in the European region as well as in the three additional regions (see Figure 3b and

methods). These additional regions include a Mediterranean region where Fig 1 indicates a general drying, a Central European region where the change from drying to wetter conditions is likely to be located, and a Northern European region which is expected to get wetter. Figure 3a shows the spread between CMIP6 models for the different terms in the regional atmospheric energy budget equation, compared with reanalysis datasets and observations over the four regions. The fluxes are only evaluated over land because of the higher societal importance compared to values over the ocean, and because E-OBS is only

available over land.

Results show that the two reanalysis datasets have the most extreme values of SW and LW radiative cooling, with ERA5 having a strongly positive and MERRA-2 a strongly negative SW radiative cooling, and MERRA-2 having a strongly positive and ERA5 having a strongly negative LW radiative cooling. The CMIP6 models' SW and LW radiative cooling are mostly located within the spread of these two reanalysis extremes. For the SH contribution to precipitation, the CMIP6 models also

mostly lie between the two reanalysis values for the large European region and the Mediterranean region, with MERRA-2 having a large radiative cooling over the Eastern Mediterranean land regions (see A2, showing energy budget maps for ERA5 and MERRA-2). For the Central European and Northern European regions the SH terms of the reanalysis data are more similar to each other, with CMIP6 models both having a stronger cooling and warming than the reanalysis datasets. For the H term, the reanalysis data is more comparable, with consistently positive contributions. MERRA-2 has higher H values than ERA5

for all except the Northern European region. The CMIP6 models have a large spread crossing the zero line, but with a positive contribution to precipitation for the model mean (as also seen in Figure 2).

With the large spread between the models for the individual energy budget terms, the resulting spread in precipitation change is large, underlining the uncertainty in how precipitation is represented in GCMs. One crucial question is the role of the evolution of sea surface temperatures, which is known to vary between models. To test the role of this to European precipitation

changes, we utilize simulations from AMIP (Atmospheric Model Intercomparison Project), which are forced with observed sea surface temperature and sea ice evolution to avoid ocean-atmosphere feedback – as such removing this inter-model uncertainty. A3 shows the change in the atmospheric energy budget components for a subset of AMIP models. Here we analyze the linear trend in each component across 1980-2014 since AMIP simulations are only available until the end of 2014. But even with this forced similarity, the models still show a large spread, indicating that the ocean does not explain the model

diversity in precipitation change, nor in the different terms of the atmospheric energy budget. The uncertainty connected to natural variability is also shown for two models; UKESM0-1-LL with high ECS, and MIROC6 with low ECS for their ensemble members. The spread between the ensemble members is smaller than the spread shown between the different models indicating that natural variability is also not a large cause of the uncertainty in the atmospheric energy budget.



## 3.2 Future Projections

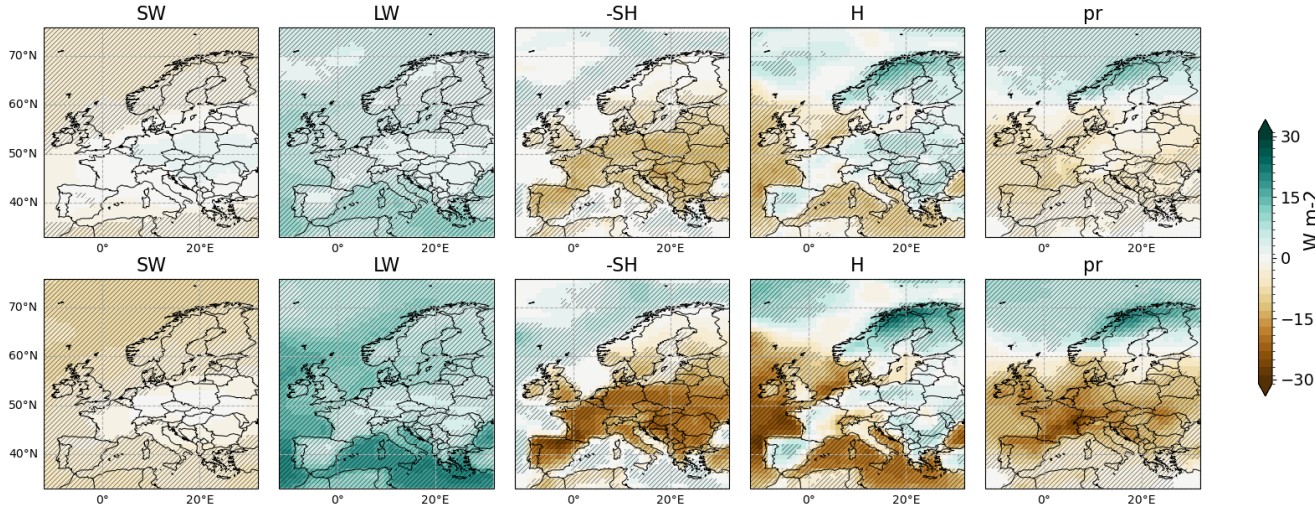

**Figure 4: The change in the summer (JJA) atmospheric energy budget from the reference period (1980-1999). Upper row shows the change at mid-century (2040 – 2059) while bottom row shows the change at end-of-century (2080 – 2099), following the SSP5-8.5 projections. Maps are averages of the 35 CMIP6 models, and hatched regions show where more than 75% of the models agree on sign of change.**

The change in future projections of precipitation is a drying over continental Europe and an increase over Northern Scandinavia, with a decreasing area where models disagree with higher forcing. The signal in change is also stronger for this high-end forcing scenario, compared to the middle of the road SSP2-4.5 (Shown in A4).

The change in forcing for future projections is clearly seen in the change in SW radiative cooling for the CMIP6 models, where at the end of the century the area where less than 75% agree on sign of change is reduced to cover the continental Europe, while the other regions show a negative contribution to precipitation increase for SW. LW radiative cooling shows a generally positive contribution to precipitation, that is increasing with time towards the end-of-the century (compare top and bottom row of Fig. 4). The change in SH and H also strengthens from mid-century to end-of-century.

The H term shows a low model agreement on the sign of change over land compared to the other terms, especially for the end-of-century change with the strongest forcing. To further understand this, the regional mean change for each model over land (similar to Figure 3) is shown in Figure 5.



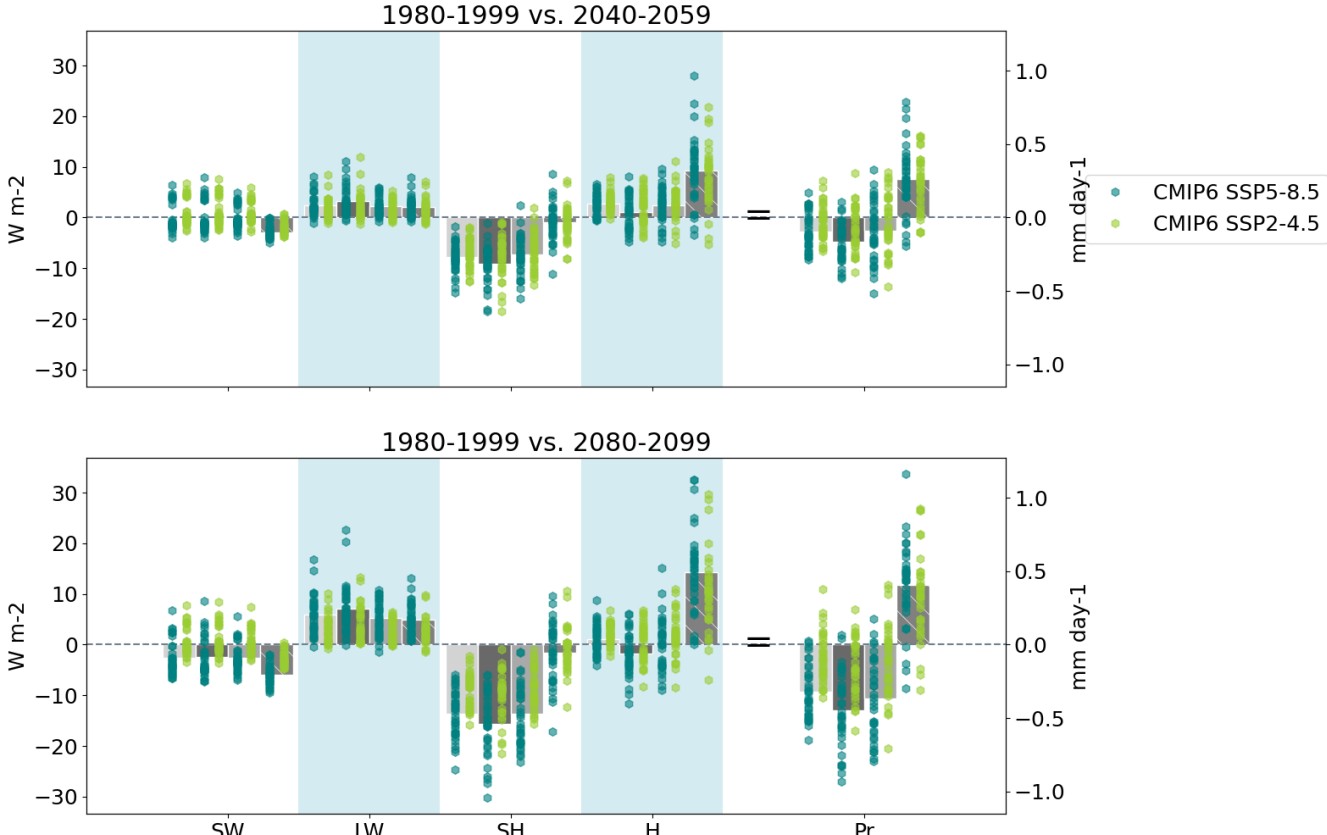

**Figure 5: The change in the summer (JJA) regional atmospheric energy budget for future projections with dark green SSP5-8.5 and light green SSP2-4.5 spatial averaged over land for the same regions as Figure 3. The bars show the mean model change for the SSP5-8.5 projections. The upper plot shows the change from 1980-1999 to middle-of-century (2040-2059) and lower show 1980-1999 to end-of-century (2080-2099).**

Results show that for all regions except the Northern European region, the H term model results are spread out around zero for both projections, for both the middle and end of century change. Thus, the contribution of dry static energy to CMIP6 precipitation change can be both positive and a negative depending on model, resulting in a model mean close to zero and also explaining the lower model agreement for the H term over continental Europe, compared to the other terms (Fig. 4). The model spread in radiative cooling from SW and LW are lowest among the energy budget terms, but especially for SW stretches across the zero line, making their future contribution to precipitation change highly model dependent. SH, on the other hand, has a larger model spread, but mostly shows a consistently negative contribution to precipitation (indicating increased SH flux) over land for all of the models, except over the Central and Northern European regions where some models show a positive contribution. Given this high model spread in the different energy budget terms, the resulting spread in projected precipitation change is substantial. It should be noted, however, that the change over the different regions gets stronger with forcing and time, even though the spread is similar for the two periods and the two forcings studied.



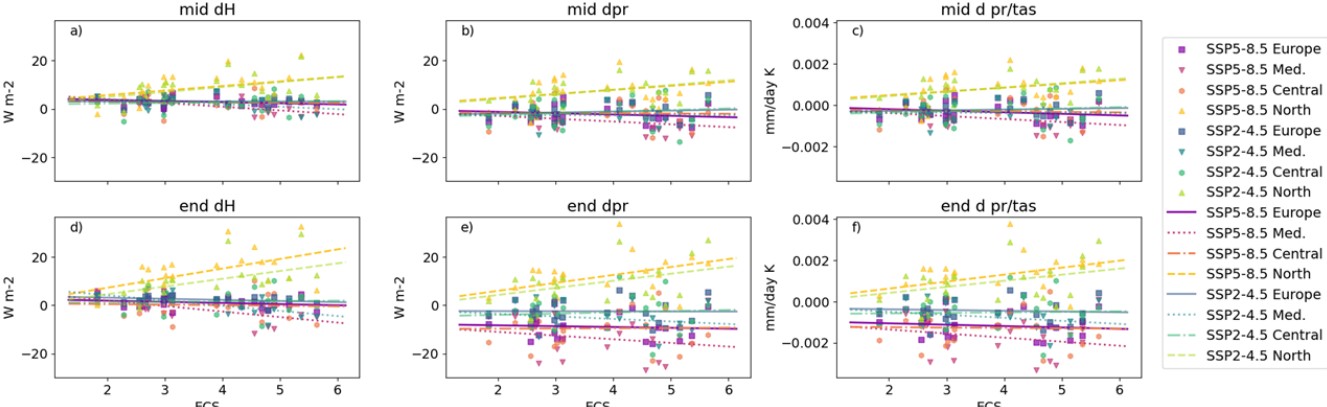

**Figure 6: Comparing the change in H (first column), precipitation (middle column) and precipitation normalized with global near surface temperature (tas) (last column) with ECS. The upper row shows the change to the middle of the century (1980-1999 vs. 2040-2059) and the row below shows the changes for the end of the century (1980-1999 vs. 2080-2099). The squares show the change over land points in the European area, down pointing triangles for Mediterranean, hexagons for the Central European region and up pointing triangles for Northern European. The lines represent the linear interpolation for the different regions with corresponding colours to the markers.**

We have used the atmospheric energy equation over Europe to explain model differences in historical precipitation change. The biggest uncertainty in sign (if the term contributes to increase or decrease in precipitation) is found for the H term. Here, we investigate the role of model variations in ECS to the dry static energy flux divergence. Figure 6 shows how the changes in the H and precipitation differ with ECS over the different regions. The linear interpolations show that models with higher ECS have a larger contribution from the H term and a larger increase in precipitation for the Northern region, with a statistical significance of ~0.5 for the H term and ~0.4 for precipitation at the end-of-century for both forcing scenarios (see B2). Over the Mediterranean there is a drying with higher ECS, however only H shows a statistically significant correlation of around 0.6 at the end-of- century for both forcings. The other regions show little change with the model ECS. To see the effect of temperature on precipitation changes, Figures 6 c and 6f show the change when precipitation change is normalized with temperature change. The results are similar to only precipitation with reduced correlations.

## 4. Discussion

The difference between the reanalysis and ESM is notable other studies have also looked at how they compare to satellite observations. Loeb *et al.* (2022) found that ERA5 shows low net TOA (top of atmosphere) flux trends compared to the CERES satellite and that the mean TOA fluxes are nine times larger than CERES over the northern hemisphere. Similar to older versions of GCMs, MERRA-2 has a too transparent atmosphere causing too little SW radiation being absorbed in the atmosphere compared to observations, while ERA5 overestimates the absorption (Wild and Bosilovich, 2024). For MERRA-2, the result is too much SW radiation and too little LW radiation at the surface. This tendency has been improved over



generations of GCMs (Wild, 2020), due to better parameterizations. However, some GCMs and MERRA-2 still have this issue, and this is also shown here. There are also other reasons for model radiation differences, e.g. CMIP6 models have different

treatment of SW scatter and absorption for aerosols. (Moseid *et al.*, 2020), in addition to different cloud parameterizations etc. Differences in SH between observations, ERA5, MERRA-2 and CMIP6 have been pointed out previously, with the reanalysis having positive global trends and CMIP6 models negative trends (Myhre *et al.*, 2024). The SH processes and parameterization differ over the models and there are large differences between the reanalysis datasets over land, highlighting the large uncertainty connected to this term.

Even though the signal in the precipitation change since 1980 is not strong in the CMIP6 models, future projections show a summer drying over Mediterranean and Central Europe, as well as more precipitation over Northen Scandinavia. This signal is strengthening with higher greenhouse gas scenario with the strongest signal for the end of century change with SSP5-8.5 and for models with high ECS. This summer drying trend is stronger, with less variability between the CMIP6 models, than what has previously been found for the CMIP5 model ensemble (Palmer, Booth and McSweeney, 2021).

The change in the dry static energy flux divergence accounts for much of the spatial variability in the models (Muller and O'Gorman, 2011), and models show both a positive and negative contribution from this term over land over continental Europe. Palmer, Booth and McSweeney, (2021) found a negative relationship between global temperature change and precipitation for central Europe and the Mediterranean, while little change in northern Europe. However, the study uses SREX regions used for EUCP (European Climate Prediction Project), while results are presented here over regions where there is expected change.

## 5. Conclusions

The CMIP6 model mean precipitation change over the historical period shows an increase in precipitation in northern Scandinavia and the Alps and a drying over southern Europe. This change is weaker than changes based on the two different reanalysis datasets as well as gridded observations, and there is considerable model disagreement. For the other terms in the regional atmospheric energy budget, there are larger areas with model agreements as well as stronger signals of change between

the two historical periods for the CMIP6 models. Comparing the atmospheric radiative cooling and the SH terms in CMIP6 to reanalysis datasets, the changes in the individual CMIP6 models fall between the two datasets, and the reanalysis results in many cases show the opposite sign of each other. This shows that there is more uncertainty in the observationally constrained reanalysis data for the atmospheric energy budget, compared to the CMIP6 models for European summer change. The large difference between the reanalysis datasets also makes it challenging to constrain the CMIP6 model results and reduce future

uncertainty.

The area where models disagree on the change from drying to wetting is reduced with a higher greenhouse gas scenario due to a larger area of drying of central Europe. Even though the models agree on sign, the spread in precipitation change between the CMIP6 models is still large. Development has been done to improve the parameterization of radiation, however more work needs to be done to understand the difference between the models for the terms with the largest spread, e.g. SH and H, in order




to understand the precipitation response. The Northern Europe region shows that the change in the H term has a positive contribution to precipitation, and models with high ECS have a larger, statistically significant, positive contribution. Similarly, models with high ECS have a larger negative contribution from H and a stronger drying over the Mediterranean. Normalizing the precipitation with temperature change in the region does not alter this result.

Further development of the reanalysis models, which assimilates observations to better represent the real atmosphere, is
necessary to enhance its use in comparing energy budgets to GCMs.

**Appendices**

Appendix A

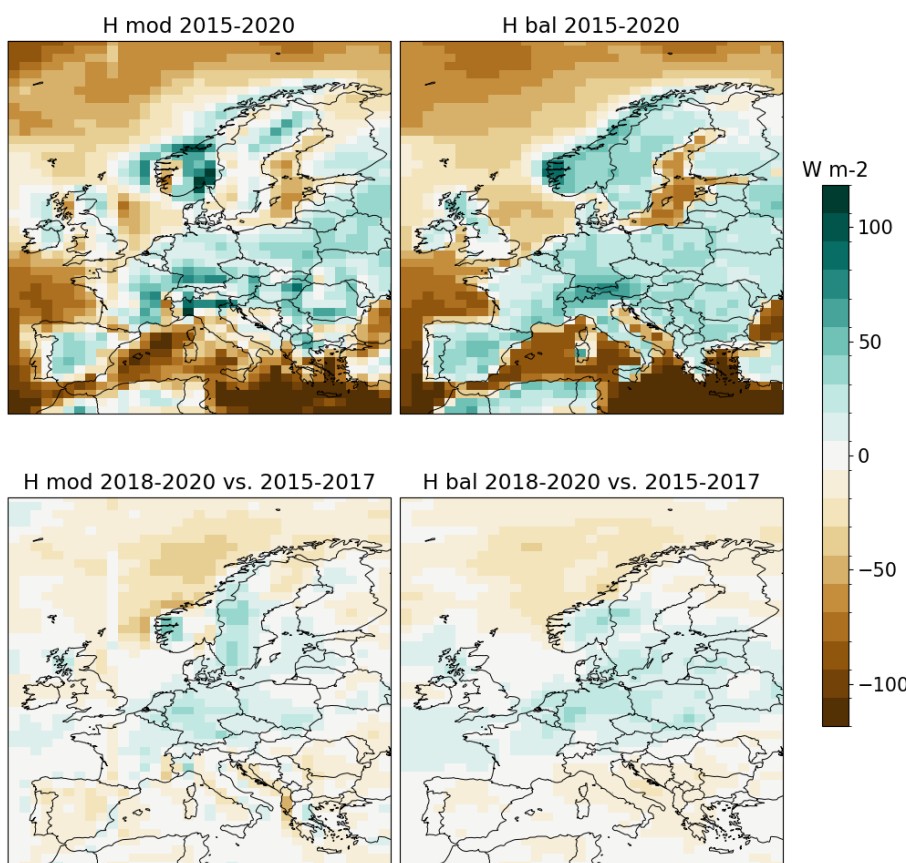

**A1: Summer (JJA) dry static energy flux divergence term (H) from vertically integrated 3-hourly fields from the NorESM2-MM**
**model (left column), and as a residual (right column). Mean over the six years 2015-2020 (top row) and the difference between the three last and the three first (bottom row).**





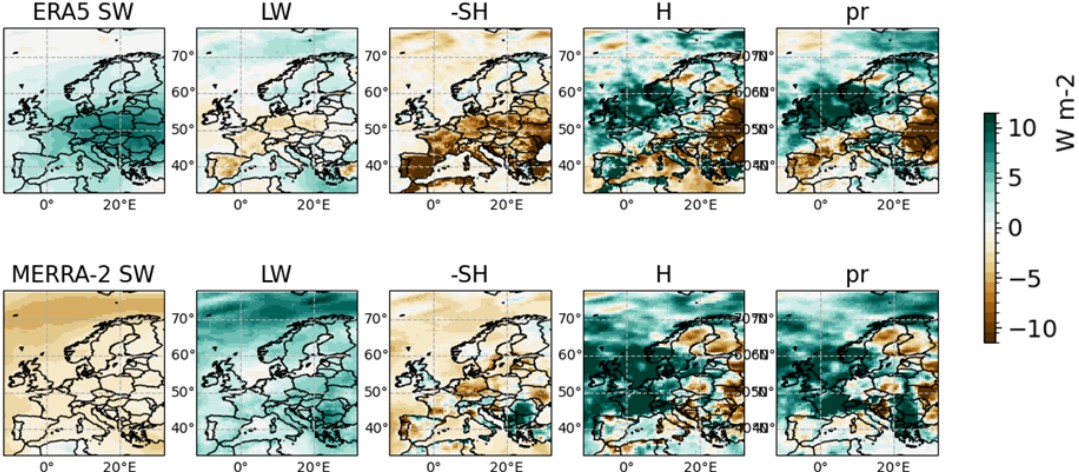


**A2: The terms of the atmospheric energy regional difference over the two 20-year periods 1980-1999 and 2004-2023 for ERA5 (upper row) and MERRA-2 (below).**

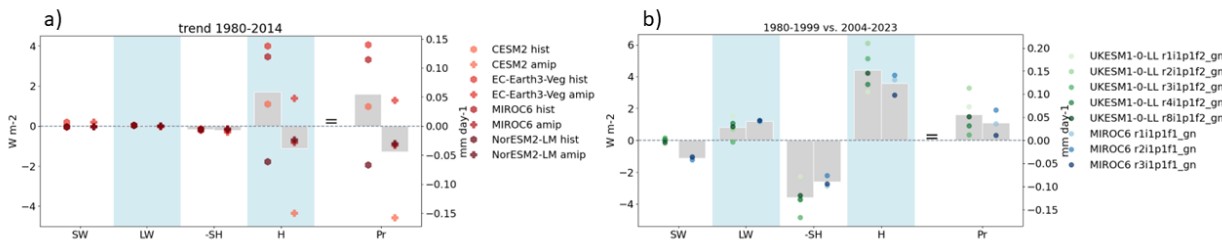

**A3: The change in the summer (JJA) regional atmospheric energy budget terms for a) the trends from 1980-2014 for AMIP**
**simulations (x) and coupled (circle), and b) the difference from 1980-1999 vs. 2004-2023 for five ensemble members of UKESM0-1-LL in different shades of green and three ensemble members of MIROC6 in blues, with the respective ensemble mean shown in bars for the larger European land region shown in Figure 3b).**





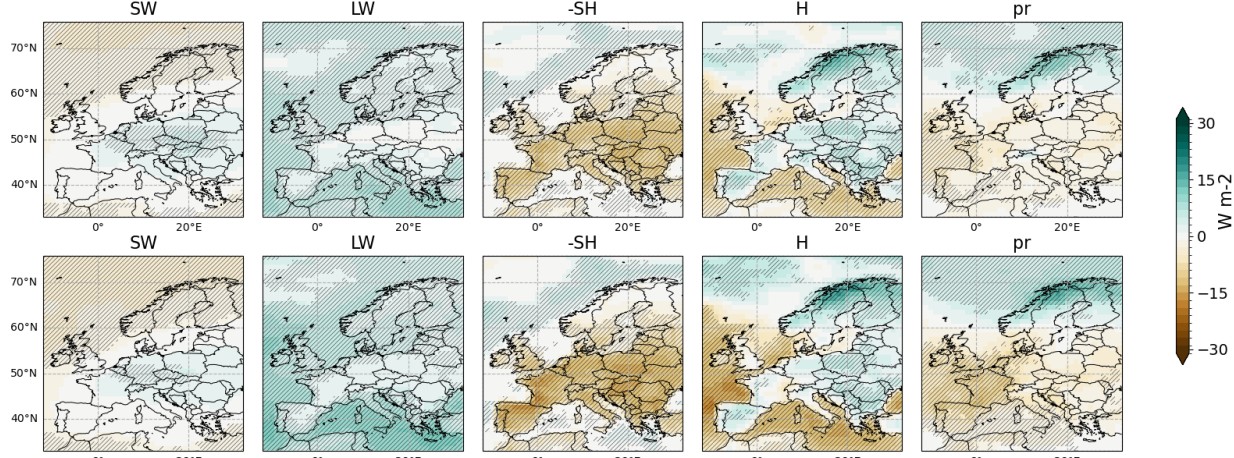

**A4: Same as Figure 4, however with SSP2-4.5 projections.**


# Appendix B

**B1. List of the models used in this study**

| Model | Resolution | Institution | Atmospheric module |
|---|---|---|---|
| ACCESS-CM2 | 199 x 144 | CSIRO (Commonwealth Scientific and Industrial Research Organisation), ARCCSS (Australian Research Council Centre of Excellence for Climate System Science) | MetUM-HadGEM3-GA7.1 |
| ACCESS-ESM1-5 | 192 x 145 | CSIRO Commonwealth Scientific and Industrial Research Organisation | HadGAM2 |
| AWI-CM-1-1-MR | 384 x 193 | Alfred Wegener Institute | ECHAM6.3.04p1 |
| BCC-CSM2_MR | 320 x 160 | Beijing Climate Center | |
| CAMS-CSM1-0 | 320 x 160 | SAMS, Chinese Academy of Meteorological Sciences | ECHAM5_CAMS |
| CAS-ESM2-0 | 256 x 128 | CAS, Chinese Academy of Sciences, | IAP AGCM 5.0 |
| CESM2 | 288 x 192 | NCAR National Center for Atmospheric Research | |
| CESM2-WACCM | 288 x 192 | NCAR National Center for Atmospheric Research | |
| CMCC-CM2-SR5 | 288 x 192 | CMCC Fondazione Centro Euro-Mediterraneo sui Cambiamenti Climatici | CAM5.3 |
| CMCC-ESM2 | 288 x 192 | CMCC Fondazione Centro Euro-Mediterraneo sui Cambiamenti Climatici | CAM5.3 |



| CNRM-CM6-1 | 256 x 128 | CNRM-CERFACS CNRM, Centre National de Recherches Meteorologiques | |
| CNRM-ESM2-1 | 256 x 128 | CNRM-CERFACS CNRM (Centre National de Recherches Meteorologiques, ), CERFACS (Centre Europeen de Recherche et de Formation Avancee en Calcul Scientifique, | |
| CanESM5 | 128 x 64 | CCCma Canadian Centre for Climate Modelling and Analysis, | CanAM5 |
| CanESM5-1 | 128 x 64 | CCCma Canadian Centre for Climate Modelling and Analysis | CanAM5.1 |
| CanESM5-CanOE | 128 x 64 | CCCma Canadian Centre for Climate Modelling and Analysis | CanAM5 |
| EC-Earth3 | 512 x 256 | EC-Earth-Consortium | IFS cy36r4 |
| EC-Earth3-Veg | 512 x 256 | EC-Earth-Consortium | IFS cy36r4 |
| FGOALS-g3 | 180 x 80 | CAS Chinese Academy of Sciences | GAMIL2 |
| GFDL-ESM4 | 288 x 180 | NOAA-GFDL National Oceanic and Atmospheric Administration, | GFDL-AM4.1 |
| GISS-E2-1-G | 144 x 90 | NASA-GISS Goddard Institute for Space Studies | GISS-E2.1 |
| GISS-E2-1-H | 144 x 90 | NASA-GISS Goddard Institute for Space Studies | GISS-E2.1 |
| GISS-E2-2-G | 144 x 90 | NASA-GISS Goddard Institute for Space Studies | GISS-E2.2 |
| IITM-ESM | 192 x 94 | Centre for Climate Change Research | IITM-GFSv1 |
| INM-CM4-8 | 180 x 120 | INM Institute for Numerical Mathematics, Russian Academy of Science | INM-AM4-8 |
| INM-CM5-0 | 180 x 120 | INM Institute for Numerical Mathematics, Russian Academy of Science | INM-AM5-0 |
| IPSL-CM6A-LR | 144 x 143 | IPSL Institut Pierre Simon Laplace | |
| MIROC-ES2L | 128 x 64 | MIROC JAMSTEC | CCSR AGCM |
| MIROC6 | 256 x 128 | MIROC JAMSTEC | CCSR AGCM |
| MPI-ESM1-2-HR | 384 x 192 | MPI-M Max Planck Institute for Meteorology | ECHAM6.3 |
| MPI-ESM1-2-LR | 192 x 96 | MPI-M Max Planck Institute for Meteorology | ECHAM6.3 |
| MRI-ESM2-0 | 320 x 160 | MRI Meteorological Research Institute, Tsukuba | MRI-AGCM3.5 |
| NorESM2-LM | 144 x 96 | NCC NorESM Climate modeling Consortium | CAM-OSLO |
| NorESM2-MM | 288 x 192 | NCC NorESM Climate modeling Consortium | CAM-OSLO |





| TaiESM1 | 288 x 192 | AS-RCEC Research Center for Environmental Changes, Academia Sinica, Nankang | TaiAM1 |
| UKESM1-0-LL | 192 x 144 | MOHC Met Office Hadley Centre | MetUM-HadGEM3-GA7.1 |

**B2. Correlation between ECS and the change in H term, precipitation and precipitation normalized with global temperature. * Statistical significance p<0.1, ** statistical significance p<=0.05.**

|  | Europe | | Mediterranean | | Central E. | | North E. | |
| --- | --- | --- | --- | --- | --- | --- | --- | --- |
| SSP: | 2-4.5 | 5-8.5 | 2-4.5 | 5-8.5 | 2-4.5 | 5-8.5 | 2-4.5 | 5-8.5 |
| H vs ECS |  |  |  |  |  |  |  |  |
| Mid century | -0.02 | -0.32 | -0.39* | -0.61** | 0.08 | -0.06 | 0.42* | 0.34 |
| End century | -0.30 | -0.24 | -0.64** | -0.56** | 0.03 | -0.05 | 0.51** | 0.52** |
| Precip vs ECS |  |  |  |  |  |  |  |  |
| Mid century | 0.13 | -0.16 | 0 | -0.29 | 0.13 | -0.03 | 0.44** | 0.30 |
| End century | -0.01 | -0.07 | -0.21 | -0.22 | 0.07 | 0.01 | 0.40* | 0.41* |
| Precip/tas vs ECS |  |  |  |  |  |  |  |  |
| Mid century | 0.2 | -0.06 | 0.08 | -0.14 | 0.22 | 0.09 | 0.19 | 0.09 |
| End century | 0.1 | 0.25 | -0.05 | 0.07 | 0.20 | 0.3 | 0.13 | 0.16 |

**Data Availability**

The MERRA2 radiation and temperature products are available at the NASA GES DISC ordering tool: https://goldsmr4.gesdisc.eosdis.nasa.gov/opendap/MERRA2_MONTHLY/M2TMNXRAD.5.12.4/, https://goldsmr4.gesdisc.eosdis.nasa.gov/opendap/MERRA2/M2T1NXSLV.5.12.4/. The ERA5 radiation and meteorological prodcuts are available at the Copernicus Climate Data Store: https://cds.climate.copernicus.eu/datasets/reanalysis-era5-single-levels-monthly-means?tab=download. Outputs from CMIP6 simulations are freely available from the CMIP6 repository on the Earth System Grid Federation nodes (https://esgf-node.llnl.gov/search/cmip6/), World Climate Research Program, 2020).

**Author Contribution**

GM and BMS conceptualized the study. BMS lead the writing, BMS and RB contributed with figures. AG contributed with dry static energy flux divergence calculations. BMS, GM, RB, CJ and CWS contributed with analysing the results, and all authors contributed with reviewing the final manuscript.



**Competing interests**

At least one of the (co-)authors is a member of the editorial board of Atmospheric Chemistry and Physics.

**Financial Support**

Authors acknowledge funding by Norwegian Research Council project BUDGET (GA number 325270).

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
