# Peer review of "European summer precipitation"

_EGUsphere, 2025_

## Author Comment (AC1)

We are thankful for the useful comments provided by the reviewers and think the strengthen out results. Review comments are presented in *italics*, with answers in blue normal font. Changes in manuscript is in *blue italic font*.

Reviewer Comments 1:

*General comments*

*This study presents changes in summertime precipitation over Europe, comparing historical simulations with reanalyses and observations, as well as future projections. It also relates them to changes in the atmospheric energy budget that sets physical constraints to the precipitation change. It is a well-motivated study, precipitation projection is highly relevant and models differ in their historical and future simulations. Applying a regional energy balance perspective is purposeful and a comprehensive comparison between CMIP6 models and two reanalysis data sets, as well as observations, is made.*

*Main conclusions are that reanalyses give a weak constraint on the energy fluxes, and that models differ even more for future projected energy fluxes and precipitation changes, but do agree on future precipitation reduction in summer over Europe. The greatest spread among the driving terms comes from the horizontal energy flux which is calculated as a residual.*

*I have some concerns regarding energy conservation in reanalysis, and the calculation of H as a residual. I would also appreciate some clarification of the formulation of the underlying energy balance equation. The authors could further strengthen their reasoning in some places, eg address more generally if/how from the model evaluation there is reason to trust the models in their future projections. They might explore more if the future projections could be constrained by skill in historical simulations of precipitation change or energy budget.*

*Regarding the presentation of results I see some shortcomings on statistical significance and use of multi-model means, and suggest some improvements to graphics presenting the results.*

*With these concerns addressed, I believe the paper should be well suited for publication in ACP.*

**Response**

Thank you for your helpful comments on our manuscript. The feedback will improve our manuscript.

*Specific comments*

*1. The framing of the atmospheric energy budget (eq 1) could use some clarification. It would help to make clear where the energy budget is calculated and each of the fluxes referred to, as there is now a mixed discussion of in-atmosphere, TOA and surface fluxes. Also the sign convention should be made clear. Does plus/minus refer to up/down or gain/loss, etc?*

*For mean precipitation what matters is the energy budget within the atmosphere, where there is a net radiative cooling (from SW absorption, and larger LW emission) compensated for by latent and to some extent sensible heat fluxes. Therefore "shortwave cooling" is a bit counterintuitive, and it is not clear why longwave cooling (dLW) and shortwave cooling (dSW) have the same sign in eq 1. Not least as the example given is an increased absorption (presumably a negative cooling) leading to decreased precipitation, and the field apparently has both negative and positive values over the domain (Fig 2). It might be helpful to present the global mean balance, SW – LW  = LH + SH  (with SW, LW, LH and SH all positive) or the regional budget for that matter SW – LW  = LH + SH + H, leading to eq 1, where the d denotes change in response to climate perturbation.*

We agree that more description on how precipitation change is linked with the regional atmospheric energy balance will improve the manuscript and have added l. 79-82:

*Globally, the atmospheric energy budget represents the balance between surface precipitation and the column integrated diabatic colling Q (excluding latent heat). Q is expressed as net longwave (LW) and shortwave (SW) radiative losses of the atmospheric column minus the net upward sensible heat (SH) flux. The radiative cooling is found by the difference between upward radiation at the top of the atmosphere (TOA) and at the surface.*

*Also, eq. 1 describes the changes in each of the fluxes (d) whereas the figures (fig 2, 3, 4,  5, …) are denoted without d.*

We are thankful for noticing this inconsistency between the equation and figures and have added delta to the figures.

*The Discussion in Section 4 on comparison with observations is largely about TOA and surface fluxes without a clear link to the in-atmosphere energy budget perspective taken in the paper.*

We agree that the TOA and surface fluxes discussed should be better described with an introduction and have added at the start of the section l. 254-255:

*The diabatic cooling Q of the atmosphere is dependent on SH at surface and the upward radiation difference between TOA and surface.*

*2. Energy conservation of the CMIP models is discussed, but do you have reason to believe that the reanalysis data sets are energy conserving? If not (and you already refer to Wild and Bosilovich 2024), can you check that and comment on if/how that affects the results? When H is calculated as a residual, all potential imbalance will be placed in this term, which might then not actually represent horizontal transport only. For the models too, this aspect of the method and potential influence on results could be more prominently discussed – currently there is a figure comparing residual and actual H for one model in the appendix. You might want to check/show other models and reanalysis, or quantify the bias between explicit and residual H, and thereby the error in estimated precipitation change.*

We agree that the energy conservation of the reanalysis should be included and have added l 108-109:

*The reanalysis models, although they aim to represent the atmosphere, are not constrained to conserve energy. As a result, the H term is not physically consistent in Equation 1. For the CMIP models that are constrained, Appendix 1…*

The analysis could benefit from calculating the direct H term for all the models, however as the term includes eddy terms, and should be evaluated at pressure levels the data from the models are not available. Further analysis on energy conservation is described in Irving et al. (2021), where they found that there is less drift from the CMIP6 models compared to CMIP5.  Although analyzing energy conservation in a regional seasonal aspect would be interesting, it is beyond the scope of this manuscript. We have added to the manuscript l 114-116.

*To calculate a direct H term with the eddy contribution, high temporal data (e.g. 3 hours used for the NorESM2 calculations) on pressure levels is needed, and this is unfortunately not available for the other CMIP6 models.*

*3. In Section 3.1 and Figure 3.1, no measure of statistical significance is given in the discussion of regionally varying trends. For the CMIP6 mean there is indication of areas where a majority of the models studied agree in sign, but this is not a robust measure, especially given the model list including multiple models more and less closely related. Even the use of an unweighted multi model mean can be questionable (see eg Kuma et al 2023 on model code genealogy). Conclusions based on where models agree are not necessarily robust.*

We change the hatched area to represent where the change is statistically significant with a two-sided student t-test p value <= 0.1 and changed text accordingly.

The study aims to analyze the spread in precipitation in the global climate models part of CMIP6 over Europe during the summer months, for historical and future projections. Rather than aiming for a definitive estimate of summer precipitation change, we use the atmospheric energy balance to explore the underlying model spread, despite the known interdependencies between models.

*You might also comment on the similarity between the two reanalysis data sets, despite the difference in assimilation of observed precipitation data.*

We added to the manuscript l181-187:

*While the H term for the reanalysis is not consistent with equation 1, since the reanalysis is not constrained to be energy conservative, they do compare better to the CMIP6 models than the other terms, with consistently positive contributions. Even with MERRA-2 that assimilates precipitation observations, there is large differences between the reanalysis and the gridded observations, except over Central Europe where the reanalysis data and observations show similar change. MERRA-2 has more mean summer precipitation in the later period for all regions compared to E-OBS, except the Northern Europe. ERA5 have the opposite change in precipitation with lower increases or decreases in the later period compared to E-OBS, and larger positive change over Northern Europe.*

*The same question regarding significance and agreement applies for section 3.2 and Figure 4.*

Changed model agreement to where the change is statistical significant with a two-sided student t-test p value <= 0.1 and changed text accordingly.

*4. Going from evaluation to future projection (Section 3.1 to 3.2) you don't really address the question of what reason we have from the model evaluation to trust the model projections. From figure 5 it seems like models agree better on future precipitation changes (in sign) than on the energy budget terms controlling it. Why is that? Assuming the energy budget framework is physically sound, the question seems to be – why do the model precipitation not adhere to it, and what is it instead that controls the precipitation change in the models?*

We agree that these points should be better described and discussed in the manuscript and have added at the end of section 3.1 l201-202:

*With the limitations and uncertainties in the historical reanalysis data, it is challenging to constrain the spread in CMIP6 models. Understanding the response to future climate projections in the different models is of importance.*

As described in the manuscript, the terms of the balance equation is based on different parameterizations in the models, where especially SH is difficult to estimate. As the H term is calculated as a residual, the balance holds for each model in Figure 5. However, the change in precipitation is regionally very different, with increases and decreases for areas where the other terms of the balance equation show equal sign. This is caused by transport, where the H term accounts for much of the spatial variability (Muller and O'Gorman, 2011).

*5. The potential for model constraint is not fully explored. Do the models that fall within the reanalysis precipitation range have smaller error included in their residual term (better agreement between balance-calculated and model-derived H)? Do the models that fall within the reanalysis range for the fluxes produce a more constrained range for precipitation than the full ensemble? Can we in fact learn something about precipitation representation in these models, from the evaluation done here? Could the historical precipitation change be applied to create a constrained future projection span, using the models that perform better (ie using Figure 3 to constrain Fig 5)?*

 It is difficult to constrain the future precipitation from the balance equation with the reanalysis data, they are too uncertain in the parameterization of SW, LW and SH shown both here over Europe in summer, and globally in Wild, and Bosilovich (2024) and Myhre et al. (2024). Even though some models fall within the precipitation range, the spread caused by natural variability makes it difficult to say that these models performs better than others. Despite the spread in historical precipitation across the models, they exhibit a similar response to global warming and climate sensitivity where Northern Scandinavia is projected to become wetter, while the Mediterranean region is projected to become drier. There is however a large spread in all the terms of the balance equation, that shows the need for better parameterizations in the models to reduce the uncertainties.

*6. The attempt to relate results to model ECS (Figure 6 and discussion) is not fully motivated. Is there a reason for the flux divergence to be related to sensitivity, or is the choice of relating H to ECS based on the large spread in H among the models? Given that H is calculated as a residual, it is even less clear why this should have a physical relation to sensitivity, and the result of low correlations is thereby not surprising. The weak relation between change in precipitation and model sensitivity would be worth commenting on in relation to previous literature, how come the general features of regional drying and wettening don't scale with model sensitivity?*

We agree that a better motivation for including the correlation between H term and model ECS and have added l 247-248.

*Since the H term accounts for much of the spatial variability of precipitation changes, ..*

*I think this section, if you want to include it, needs some more explanation and elaboration. Currently the statement on L257-258 that the signal strengthens with higher ECS seems a bit strong. You might also want to relate to studies like Barnes et al (2024) who look closer at regional climate projection and to what extent it can be attributed to model climate sensitivity,*

We agree that more comparisons and explanation are needed, and have added 276-277:

*Although the response to the future forcing scenarios is stronger in models with high ECS, the response to the forcings with a drying south and wetter north with time and forcing is found across the models.*

The temperature response in regions to ESC is not strictly comparable to the precipitation response, as seen with the greater uncertainty in precipitation response to global warming compared to precipitation. The results found in Barnes et al. (2024) is therefore difficult to compare with this study, as the results were found from observationally constrained signals. However, future analysis should be done to try to constrain the hydrological signal to climate change with machine learning.

*7. For figure presentation, please add measures of statistical significance where possible, and please make figures and fonts larger (especially figures 2 and 6). In Figure 3, the symbols for the two reanalyses are inseparable, as are the differently shaded grey bars. It would help to choose these symbols and colours differently. The = sign in the chart is a bit confusingly placed, and would perhaps fit better in the x-label if you want to include it. See comment above regarding d, and sign convention for the terms.*

We have gone through the Figures, and added statistical significance and tried to make improvement according to the suggestions.

*Technical corrections*

*L13 compares -> compare*

Changed accordingly

*L14 in the atmospheric energy **budget***

Changed accordingly

*L15 "diversity" has positive connotation, when what is actually described is models deviating from each other and thereby from observations. Maybe spread is a more neutral word*

Reworded the sentence to:

*In future projections, variability across models increases relative to the historical period for the changes in precipitation, sensible heat and the dry static energy flux divergence, highlighting the uncertainties in the magnitude of these terms.*

*L33 each degree **of** warming*

Changed accordingly

*L87 (DeAngelis, … Wild) references repeated*

Fixed, thank you for noticing.

*L91 this is pretty much a textbook statement regarding radiative effects of clouds, so the choice of references seems a bit arbitrary*

Rewrote the sentence:

*The physical representation and geographical location of clouds is found to be critically important for whether they contribute to radiative cooling or heating found in both observations and models increasing the climate sensitivity*

*L107 monthly files, please phrase more specifically*

Rewrote to explain better:

*.. the large-scale differences in H between the model calculations and the residual from monthly mean output that is available from the CMIP6 models are comparable…*

*L242 is notable other studies -> is notable. Other studies*

Changed the sentence:

*and ESMs is notable for radiation and SH, and other studies have*

*L242 "they" -> models?*

Added :

*The terms*

*L277 **over** central Europe*

Changed accordingly

References

Irving, D. *et al*. (2021) "A Mass and Energy Conservation Analysis of Drift in the CMIP6 Ensemble," *Journal of Climate*, 34(8), pp. 3157–3170. Available at: https://doi.org/10.1175/JCLI-D-20-0281.1.

Muller, C.J. and O'Gorman, P.A. (2011) "An energetic perspective on the regional response of precipitation to climate change," *Nature Climate Change*, 1(5), pp. 266–271. Available at: https://doi.org/10.1038/nclimate1169.

Myhre, G. *et al*. (2024) "Strong contribution from sensible heat to global precipitation increase in climate models is not supported by observational based data," *Frontiers in Climate*, 6, p. 1383337. Available at: https://doi.org/10.3389/fclim.2024.1383337.

Wild, M. and Bosilovich, M.G. (2024) "The Global Energy Balance as Represented in Atmospheric Reanalyses," *Surveys in Geophysics*, 45(6), pp. 1799–1825. Available at: https://doi.org/10.1007/s10712-024-09861-9.

RC2:

*Review of "European summer precipitation" by Birthe Marie Steensen, Gunnar Myhre, Racheal Byrom, Ada Gjermundsen, Caroline Jouan, and Camilla Weum Stjern*

*General comments*

*The manuscript considers changes in summer precipitation over Europe and analyzes these changes using a regional atmospheric energy budget framework. It compares historical simulations in 35 models from CMIP6 to both observations (E-OBS daily gridded observational dataset) and reanalysis datasets (MERRA2 and ERA5), finding that the reanalysis datasets generally match observed rainfall but also exhibit large discrepancies with CMIP6 models when considering other terms in the atmospheric energy budget. Models generally agree that European summer rainfall will decrease with anthropogenic forcing.*

*Overall, the manuscript is strong, and the regional atmospheric energy budget framework which the authors apply is both sound and insightful. There are a few changes which might strengthen the manuscript which I hope the authors will consider and which, once addressed, would make the manuscript suitable for publication in ACP.*

**Response**

Thank you for reading our manuscript and your helpful comments. The feedback will improve our manuscript.

*Specific comments*

- *In the atmospheric energy budget framework, the horizontal energy flux H is calculated as a residual. While the manuscript validates this approach for the NorESM2-MM model in Appendix A1, this approach is not validated in other models (nor the reanalysis datasets) in the same way. Of any term in the atmospheric energy budget, delta H appears to be the main driver of changes in precipitation, so, if not overly cumbersome, it would be helpful to further validate the manuscript's treatment of H as a residual.*

We agree that calculating the H term directly from the models would be beneficial for the analysis, however it is an impossible task to calculate for all the models. As the term includes eddy terms and should be evaluated at pressure levels these data from the models are not available. Further analysis on energy conservation is described in Irving et al. (2021), where they found that there is less drift from the CMIP6 models compared to CMIP5. Although analyzing energy conservation in a regional seasonal aspect would be interesting, it is beyond the scope of this manuscript. We have added to the manuscript l114-116.

*To calculate a direct H term with the eddy contribution, high temporal data (e.g. 3 hours used for the NorESM2 calculations) on pressure levels is needed, and this is unfortunately not available for the other CMIP6 models.*

- *It is unclear why delta H is compared to ECS in Figure 6. While these quantities are statistically related, one would assume there is some physical reasoning which connects these quantities in a more mechanistic manner as well. If such a connection exists, the manuscript would benefit from making this connection more explicit; if not, it would benefit from clarifying that these are only correlations (and thus may be spurious).*

We agree that including the motivation for including the H term is necessary, and have added l 247-248:

*Since the H term accounts for much of the spatial variability of precipitation changes, ..*

- *The labels in several plots are difficult to read and the color choices for several quantities are difficult to distinguish (e.g., gray bars/regions in Figure 3). Consider increasing the font size of particularly small labels and changing especially similar colors/markings to be more easily distinguishable.*

We have redone the plots to improve the labels, correct delta and put black lines around the grey bars so the show better.

*Technical corrections*

- *Figures 2-6 should label each term in the atmospheric energy budget as the change in that term (e.g., delta SH) not the term itself (e.g., SH).*

We have added delta where appropriate to the figures.

- *Change to "compares" on line 13*

Changed accordingly

- *Add "budget" at the end of sentence on line 14*

Changed accordingly

- *Missing period on line 242*

Changed the sentence:

*.. and ESMs is notable for radiation and SH, and other studies have..*

- *Replace "being" with "to be" in line 245*

Changed accordingly

- *Replace "strengthening with" with "stronger in" in line 257*

Changed accordingly

- *Replace "while" with "with" in line 263*

Changed accordingly